# Prevalence of Common Viral Skin Infections in Beach Volleyball Athletes

**DOI:** 10.3390/v13112107

**Published:** 2021-10-20

**Authors:** Niki Tertipi, Vasiliki Kefala, Effie Papageorgiou, Efstathios Rallis

**Affiliations:** Department of Biomedical Sciences, School of Health Sciences, University of West Attica, GR-12243 Athens, Greece; valiakef@uniwa.gr (V.K.); effipapag@uniwa.gr (E.P.); erallis@uniwa.gr (E.R.)

**Keywords:** viral skin infections, beach volleyball athletes, epidemiology

## Abstract

Viral skin infections often affect the sports community. The aim of this study was to assess the rates, location sites, and seasons of appearance of common viral cutaneous diseases in beach volleyball athletes in Greece. Five hundred and forty-nine beach volleyball athletes participated in this study. The average age was 28.4 years. The viral infections were herpes simplex (type 1), molluscum contagiosum and warts. The measured parameters included: gender, age, the season when athletes may be more susceptible to infections and the location of infection in the body. Practicing information such as the number of training years, number of weekly trainings, and average hours of daily training was also recorded. Incidence rates correlated in relation to age: (a) warts (*p* < 0.001), molluscum contagiosum (*p* < 0.001), and herpes simplex (*p* = 0.001); (b) years of training: warts (*p* < 0.001), molluscum contagiosum (*p* < 0.001), and herpes simplex (*p* = 0.004); (c) average hours of daily training: molluscum contagiosum (*p* = 0.006) and herpes simplex (*p* < 0.010). The skin is the largest organ, and the risk of infection should not be underestimated. Prevention, early detection, recognition, and treatment are related to health and athletic performance, but also to the risk of transmission.

## 1. Introduction

Injuries that are associated with sports-related infectious diseases cause significant morbidity in athletes [1,2]. Previous studies have highlighted the contagious nature of skin infections among athletes [2]. The three main viral cutaneous infections that affect athletes are warts, molluscum contagiosum and herpes simplex [3]. Viral skin infections can be detrimental in terms of lost playing-time and potentially affect the successful careers of athletes [4]. Beach volleyball is a very popular sport among young adults and has a high injury rate [4]. As with any sport activity, beach volleyball carries a serious risk of skin infection. Injuries of the skin from cuts, abrasions or lacerations, exposure to infectious and environmental factors, physical contact between athletes, persistent sweating of athletes and sharing of equipment and facilities increase the likelihood of infection. Exposure to sunlight and tight clothes may form a perfect environment for viral skin infections [5]. The onset of skin infections in athletes is often due to the training environment. Injuries related to physical activity can alter the barrier of the skin and mucosa and allow the invasion of the infectious agent [6]. The sand contains pathogenic microbes that affect human health and can lead to serious infections [7].

The risk of infection, during training with a partner with active lesions, was estimated at 30% at contact points, such as the head, neck and extremities of the limbs. Eye damage is also common [1,8]. Molluscum contagiosum can be seen in any sport, usually observed in the athletes’ forearms, faces and hands; it spreads in beach volleyball athletes through the use of shared objects, such as a ball, and skin–skin contact [9]. Warts are caused by the human papillomavirus (HPV), which has more than 150 types. The lesions are spread by direct skin–skin contact, through shared objects, shower floors and sand [9]. Skin trauma, foot sweating and moisture are also considered to be predisposing factors for the development of warts [9]. They are usually located on hands and soles of the feet, and can be painful, especially in the feet, and may hinder the athletes’ performances. Football players, rowing athletes, wrestlers, gymnasts and swimmers can all develop warts [8].

There is no previous descriptive epidemiological research focusing exclusively on viral skin infections in beach volleyball. The existing literature is often limited to studies and does not feature recent data. Preventing the spread of infection is the second most important aspect of treatment following pharmacotherapy. The protection against infection and transmission of pathogens in sports teams is crucial to avoid unnecessary morbidity and minimize disruption to the training and competition schedule. The location point of the infections will be discussed.

## 2. Materials and Methods

In this study, an online questionnaire was created and disseminated using Google Forms. Before the initiation of the online survey, a pilot study was conducted to ensure the correctness and understanding of the questions. This study was approved by the Ethics Committee of Research of the University of West Attica (20.07.2020/app.no.48944) and by the Hellenic Volleyball Federation (06.03.2019/app.no.746).

The content of the questionnaire and the user-data collection methods ensured the privacy and dignity of the participants. Informed consent or assent was obtained from the participants. Due to the pandemic restriction measures, the link to the Google Forms questionnaire was sent via e-mail to the athletes by the Hellenic Volleyball Federation in November 2020. The questionnaire included demographic characteristics, gender, and age, as well as questions about: (a) viral skin infections—herpes simplex (type 1), molluscum contagiosum and warts; (b) the season when the appearance of the infections occurred; (c) the location on the body where these infections appeared; (d) practice information—yearly, weekly and daily practice, related to the type of infection.

### Statistical and Data Analysis

All the data collected through the questionnaire were entered into an electronic database created by Excel software. Data analysis was performed using IBM SPSS Statistics for Windows, version 26.0. Categorical variables are presented as absolute (n) and relative (%) frequencies and continuous variables are presented as means (standard deviation, SD) and/or medians (interquartile range, IR). The normality assumption was evaluated using the Kolmogorov–Smirnov criterion (*p* > 0.05 for all variables), histograms and normal probability plots. Bivariate analyses were conducted and included the X^2^ test and the X^2^ trend test to determine the associations between categorical variables, and the Mann–Whitney test to investigate group differences within continuous variables. A two-sided *p*-value of 0.05 was considered statistically significant.

## 3. Results

### 3.1. Demographic Characteristics and Training Elements

The study population consisted of 549 beach volleyball athletes. There were no invalid questionnaires. Demographic characteristics and data regarding the training of the participants in the study are presented in Table 1.

In total, 218 athletes were male, and the average age of all athletes in the study was 28.4 years (SD = 10.3). More than four out of ten (41.9%) athletes had less than 3 years of training in beach volleyball, 24.2% from 4 to 6 years, 12.7% from 10 to 12 years, 11% from 7 to 9 years and 10.2% had more than 12 years of experience. Almost four out of ten (38.4%) were training from 3 to 4 times weekly, 32.2% from 4 to 5 times, 16% more than 6 times and 13.3% from 1 to 2 times. Finally, 65.6% were training from 1 to 2 h on a daily basis, 32.2% more than 2 h and 2.2% less than 1 h.

### 3.2. Dermatological Diseases/Viral Infections, Season of Appearance and Body Location Are Presented in Below Table

In total, 106 athletes (19.3%) of the study population had been diagnosed with herpes simplex (type 1), 110 athletes (20%) with molluscum contagiosum and 153 athletes (27.9%) with warts. With regard to the body location sites, for athletes who had been diagnosed with herpes simplex, the most frequently identified body location site was their corpus (6.6%), for athletes who had been diagnosed with molluscum contagiosum, the most frequently identified body location site was their corpus (5.1%), and for athletes who had been diagnosed with warts, the most frequently identified body location site was the soles of their feet (7.5%) (see Table 2).

### 3.3. Correlations

Bivariate analyses were conducted (Table 3 and Table 4) using herpes simplex, molluscum contagiosum and warts as the dependent variables and the demographics, number of training years, number of weekly trainings and average hours of daily training as the independent variables.

Athletes who had been diagnosed with herpes were at a greater percentage male (*p* < 0.001), had a higher median age (*p* = 0.001), more years of training in beach volleyball (*p* = 0.004) and more training sessions per week (*p* = 0.010) than athletes who had not been diagnosed with herpes. Additionally, athletes who had been diagnosed with molluscum contagiosum were disproportionately male (*p* < 0.001), had a higher median age (*p* < 0.001), more years of training in beach volleyball (*p* < 0.001) and more hours of daily training (*p* = 0.006) than athletes who had not been diagnosed with molluscum contagiosum.

Finally, athletes who had been diagnosed with warts were disproportionately male (*p* < 0.001), had a higher median age (*p* < 0.001) and more years of training in beach volleyball (*p* < 0.001) than athletes who had not been diagnosed with warts.

## 4. Discussion

Viral skin infections have become more common as the number of athletes has increased [1]. To our knowledge, this is the first study to present the epidemiology of skin infections among beach volleyball athletes. The previous literature on skin infections of beach volleyball athletes has been limited to case studies [7,10,11] or broader descriptive studies involving mainly musculoskeletal injuries [2,4,6,12,13]. The main findings of this study are the rates of appearance of viral skin infections and the specific body locations where they occur. There is a lack of relevant literature which concerns the research of beach volleyball athletes, so when discussing rates of viral skin infections, the comparison between studies is difficult. In other studies, the most common sports infections that have been reported were fungal, bacterial, contact dermatitis and viral infections [1,14]. In our study, we researched the likelihood of viral skin infections being related to age, gender, the season of appearance, the location of the lesion, the number of years of experience, the weekly training schedule, and the average hours of daily training. These elements provide an examination of recent skin infection data from beach volleyball athletes, which can help to inform on the occurrence of skin infections. We studied three viral skin infections, herpes simplex (type 1), molluscum contagiosum and warts, the latter of which was the most frequent viral skin infection that occurred.

Human papilloma virus infection causes warts [15]. Approximately 5% to 30% of children and young adults who participate in wrestling will have warts [16,17]. In beach volleyball, trauma to the skin and dampness from foot perspiration may increase the susceptibility to warts [18]. In our study, males of ages over 28 years old who practiced frequently were more sensitive to the development of warts. Exercise increases body temperature and, in addition to the contact with other players, creates the right environment for the growth of warts [18]. In summer and autumn we noticed an increase in infections. Commonly, these lesions are seen on the soles of the feet and hands, but they can occur on other skin surfaces. In our study, the most frequent location site of warts was the soles of the feet, followed by lower limbs, upper limbs and hands. Warts on the soles of the feet impede performance, are occasionally painful, and they should be treated [19]. Swimmers, wrestlers and football players can all develop warts, but their appearance may be affected by the type of sport [20,21,22].

Molluscum contagiosum is an irritating viral infection and spreads through skin–lesion contact with an infected person [3,5]. It predominately affects and is most commonly seen among young children; however, it may also affect adolescents and adults [9]. Molluscum contagiosum may be contracted by adults as a sexually transmitted infection [8]. In our study, the proportion of infected males was higher than females, while other researchers observed no evidence of a difference in prevalence by gender [23]. Similarly, the risk of occurrence in beach volleyball athletes is increased, such as in those with low-frequency beach volleyball training. However, high-intensity activities cause an increase in susceptibility to infections [24,25,26]. In beach volleyball, wet sand and balls are vectors of pathogens [7]. Molluscum contagiosum is more common in wrestlers, gymnasts and swimmers [19,27]. The use of shared equipment by someone infected with molluscum contagiosum could spread the infection more widely, both to adults and children. The areas of exposed skin tend to become infected [19]. In our study, molluscum contagiosum is the second most virulent skin infection in terms of rates of infection, and summer sees increased cases. Climate is often described as being a factor associated with an increase in molluscum contagiosum infections. The highest rates that have been observed were in hot climates [28,29]. Lesions are commonly seen on the face, forearms and hands [26]. Swimmers may be particularly susceptible to infections of the soles of the feet [30]. In our study, the highest rates of body location areas were the corpus, upper and lower limbs, and none in the soles of the feet.

Herpes simplex virus (type 1) is a common infectious agent in humans. In our study, herpes incidence rates differ slightly from molluscum contagiosum. Herpes virus can cause primary and recurrent infections in athletes [8]. In wrestling, the most common viral skin infection reported was herpes [31]. The activation can be triggered by UV radiation, causing the immune system suppression and reactivation of the herpes virus. In our study, those who had developed herpes virus were predominately male, and the frequency of practice affected the appearance of the virus. Beach volleyball is a strenuous activity which involves exposure to sunlight and athletes are often training for hours, with a high risk of infection. Athletes who are exposed to increased ultraviolet solar radiation, such as snow skiers, are affected by these lesions [32]. Most cases in our survey were recorded in the summer and spring. Ground surfaces may also have a major role in the occurrence of herpes [1]. When athletes are infected by herpes, the result is a massive outbreak athletic during training. In the literature, the appearance of lesions on the face is the most common (up to 70%) [19]. In our findings, herpes lesions primarily developed on the corpus and face, and less on the head and lower limbs. The skin condition can also increase the risk of herpes skin infection. Missed athletic participation and morbidity can result from infection by herpes simplex [33,34].

Our data did not include information on the athletes’ hygiene practices. We suspect that not practicing careful individual hygiene may worsen athletes’ susceptibility to viral skin infections. Ensuring the proper environmental conditions and maintaining athletes’ hygiene can help reduce the transmission from athlete to athlete. In addition, our data did not include information of the time lost from training due to the infections. Considering these constraints, a study specifically involving the frequency and relative timing of infections is warranted.

## 5. Conclusions

Our findings may suggest that persisting viral skin infections afflict beach volleyball athletes. The incidence rates of herpes simplex, molluscum contagiosum and warts are heightened in beach volleyball athletes, especially in summer. Removing athletes from training, early diagnosis, monitoring and treatment are important methods for controlling infection until athletes are no longer contagious. Returning to play and the prevention of infection are important among athletes. Adequate knowledge and the rapid initiation of therapy helps to ensure no disruption of team exercise and competitions.

## Figures and Tables

**Table 1 viruses-13-02107-t001:** Demographic characteristics and data regarding athletes’ training.

	n	%
Gender		
Male	218	39.7
Female	331	60.3
Age	28.4 ^a^	10.3 ^b^
Number of training years		
Less than 3	221	41.9
4–6	128	24.2
7–9	58	11.0
10–12	67	12.7
More than 12	54	10.2
Number of weekly training		
1–2	73	13.3
3–4	211	38.4
4–5	177	32.2
More than 6	88	16.0
Average hours of daily training		
Less than 1	12	2.2
1–2	360	65.6
More than 2	177	32.2

Values are presented as n (absolute) and % (relative) frequencies, unless stated otherwise. ^a^ Mean value; ^b^ standard deviation (SD).

**Table 2 viruses-13-02107-t002:** Viral Infections.

		Viral Infections	
	Herpes Simplex (Type 1) n (%)	Molluscum Contagiosum n (%)	Warts n (%)
No	443 (80.7)	439 (80)	396 (72.1)
Yes	106 (19.3)	110 (20)	153 (27.9)
Season of infections’ appearance			
Winter	5 (0.9)	8 (1.5)	19 (3.5)
Spring	15 (2.7)	17 (3.1)	17 (3.1)
Summer	12 (2.2)	54 (9.8)	58 (10.6)
Autumn	11 (2.0)	19 (3.5)	25 (4.6)
Body location			
Upper limbs	0	27 (4.9%)	18 (3.3%)
Lower limbs	2 (0.4%)	13 (2.4%)	20 (3.6%)
Head	2 (0.4%)	0	1 (0.2%)
Corpus	36 (6.6%)	28 (5.1%)	8 (1.5%)
Palms/Hand	0	3 (0.5%)	18 (3.3%)
Feet-Soles	0	0	41 (7.5%)
Face	4 (0.7%)	5 (0.9%)	2 (0.4%)

Values are expressed as n (absolute) and % (relative) frequencies.

**Table 3 viruses-13-02107-t003:** Bivariate analyses using herpes and molluscum contagiosum as dependent variables.

	Herpes Simplex (Type 1)			Molluscum Contagiosum		
Characteristic	No	Yes	*p* Value	No	Yes	*p* Value
Gender			**<0.001** ^a^			**<0.001** ^a^
Male	156 (35.2)	62 (58.5)		151 (34.4)	67 (60.9)	
Female	287 (64.8)	44 (41.5)		288 (65.6)	43 (39.1)	
Age ^b^	25.0 (16)	30.0 (10)	**0.001** ^c^	25.0 (16)	30.5 (9)	**<0.001** ^c^
Number of training years			**0.004** ^d^			**<0.001** ^d^
Less than 3	193 (45.5)	28 (26.9)		199 (47.4)	22 (20.4)	
4–6	99 (23.3)	29 (27.9)		98 (23.3)	30 (27.8)	
7–9	42 (9.9)	16 (15.4)		47 (11.2)	11 (10.2)	
10–12	48 (11.3)	19 (18.3)		34 (8.1)	33 (30.6)	
More than 12	42 (9.9)	12 (11.5)		42 (10.0)	12 (11.1)	
Number of weekly trainings			**0.010 ^d^**			0.204 ^d^
1–2	64 (14.4)	9 (8.5)		62 (14.1)	11 (10.0)	
3–4	174 (39.3)	37 (34.9)		170 (38.7)	41 (37.3)	
4–5	142 (32.1)	35 (33.0)		139 (31.7)	38 (34.5)	
More than 6	63 (14.2)	25 (23.6)		68 (15.5)	20 (18.2)	
Average hours of daily training			0.540 ^d^			**0.006** ^d^
Less than 1	11 (2.5)	1 (0.9)		11 (2.5)	1 (0.9)	
1–2	285 (64.3)	75 (70.8)		298 (67.9)	62 (56.4)	
More than 2	147 (33.2)	30 (28.3)		130 (29.6)	47 (42.7)	

Values are expressed as n (%), unless stated otherwise. ^a^ X^2^ test. ^b^ Median value (IR). ^c^ Mann–Whitney test. ^d^ X^2^ test for trend.

**Table 4 viruses-13-02107-t004:** Bivariate analyses using warts as dependent variables.

	Warts		
Characteristic	No	Yes	*p* Value
Gender			<0.001 ^a^
Male	129 (32.6)	89 (58.2)	
Female	267 (67.4)	64 (41.8)	
Age ^b^	25.0 (16)	30.0 (11)	**0.001** ^c^
Number of training years			**0.004** ^d^
Less than 3	181 (47.9)	40 (26.7)	
4–6	89 (23.5)	39 (26.0)	
7–9	41 (10.8)	17 (11.3)	
10–12	29 (7.7)	38 (25.3)	
More than 12	38 (10.1)	16 (10.7)	
Number of weekly trainings			0.300 ^d^
1–2	55 (13.9)	18 (11.8)	
3–4	153 (38.6)	58 (37.9)	
4–5	129 (32.6)	48 (31.4)	
More than 6	59 (14.9)	29 (19.0)	
Average hours of daily training			0.344 ^d^
Less than 1	11 (2.8)	1 (0.7)	
1–2	260 (65.7)	100 (65.4)	
More than 2	125 (31.6)	52 (34.0)	

Values are expressed as n (%), unless stated otherwise. ^a^ X^2^ test. ^b^ Median value (IR). ^c^ Mann–Whitney test. ^d^ X^2^ test for trend.

## Data Availability

Not applicable.

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
