# Peer review of "Prevalence of Common Viral Skin Infections in Beach Volleyball Athletes"

_viruses, 2021, doi:10.3390/v13112107_

Round 1

Reviewer 1 Report

  1. In this study, 549 beach volleyball athletes participated the survey, What is criteria for the online questionnaire? How many invalid questionnaires?
  1. Can herpes simplex, molluscum 13contagiosum and warts represent “common viral skin infections”?
  2. Herpes simplex is type 1 or type 2, or both ?

Reviewer 2 Report

Thank you for your original contribution. Some remarks follow hereafter:

  1. Reference 18 is the same as references 27 and 34 and is written incorrectly in all three instances.
  2. Reference 4: “Weesner, T” instead of “Weeser, T” and “32(4), 235-237” instead of “32(4)”.
  3. Reference 9: The publication year is 2012.
  4. Reference 10: ”Cohen, PR” instead of “Muacevic, A; Adler, R.J” and please add “Cureus 2019, 11(12):e6429”.
  5. Reference 12: The journal name is the “Br J Sports Med”.
  6. Reference 13: “Paradise SL, Hu YE” instead of “Scott LP, Yao-Wen EH” and also “Curr Sports Med Rep 2021, 20(2), 92-103”.
  7. Reference 14: “Adams BB” instead of “Brian BA”.
  8. Reference 15: Please add “et al” after the authors’ names.
  9. Reference 17: “…Niedfeldt, MW et al”
  10. Reference 32: The year of publication is 2000.
  11. There are a few more typographical errors both in the main text and the references.
